

# Systemic immune-inflammation index as an independent risk factor for diabetic nephropathy: a retrospective, single-center study

Xiaohong Zhang[*], Yuan Fang[*], Mengjie Weng, Caiming Chen, Yanfang Xu and Jianxin Wan

[1] Department of Nephrology, Blood Purification Research Center, the First Affiliated Hospital, Fujian Medical University, Fuzhou, Fujian, China
[2] Fujian Clinical Research Center for Metabolic Chronic Kidney Disease, the First Affiliated Hospital, Fujian Medical University, Fuzhou, Fujian, China
[3] Department of Nephrology, National Regional Medical Center, Binhai Campus of the First Affiliated Hospital, Fujian Medical University, Fuzhou, Fujian, China
[*] These authors contributed equally to this work.

Corresponding authors
Yanfang Xu, xuyanfang99@hotmail.com
Jianxin Wan, wanjx@fjmu.edu.cn

## ABSTRACT

**Purpose.** Systemic immune-inflammation index (SII) was an indicator which could reflect immune response and systemic inflammation. We aim to explore the relationship between SII and diabetic nephropathy (DN).

**Methods.** SII was calculated as neutrophil count × platelet count/lymphocyte count. The receiver operating characteristic (ROC) curve analysis was performed to evaluate the diagnosis power of SII. Univariate and multivariate logistic analysis were conducted to assess SII as the risk factor for DN. A restricted cubic spline model was carried out to show the nonlinear association between SII and DN.

**Results.** Two hundred participants were enrolled, with an average age of $56.6 \pm 13.4$ years; 54% participants were categorized as DN. Spearman association analysis showed SII was positive associated with increased urinary albumin to creatinine ratio and serum creatinine, while negative associated with eGFR and serum albumin. The ROC curve revealed that the maximum area under the curve (AUC) was 0.761 (95% $CI$, 0.694-0.828, $P < 0.001$). After univariate and multivariate logistic analysis, SII ($OR=1.004$, $P = 0.002$) and serum creatinine ($OR=1.146$, $P < 0.001$) were risk factors for the occurrence of DN, while age ($OR=0.920$, $P = 0.011$) and serum albumin ($OR=0.708$, $P < 0.001$) were protective factors for the occurrence of DN. The restricted cubic spline model showed that there was a significant nonlinear association between DN incidence and continuous SII value when it exceeded 624*10^9/L.

**Conclusion.** SII is a novel diagnostic biomarker which is independently associated with DN. Further large-scale prospective studies are needed to validate our findings before SII can be considered a reliable diagnostic or prognostic tool for DN.

## INTRODUCTION

Diabetic nephropathy (DN) is the main cause of end-stage renal disease (*Ruiz-Ortega et al., 2020*), causing heavy medical burden worldwide. About 35–40% of patients with type 2 diabetes mellitus (T2DM) will go on to develop DN (*GBD Chronic Kidney Disease Collaboration, 2020*). The mechanisms leading to the development and progression of DN are not well understood, but inflammation and immune response appear to be important relevant factors (*Donath, 2019*; *Gnudi, Coward & Long, 2016*; *Pérez-Morales et al., 2019*).

The systemic immune-inflammation index (SII) was first developed by *Hu et al. (2014)*, and was reported to be a good indicator that could reflect the immune response and systemic inflammation in the human body. It contained three types of inflammatory cells, including neutrophil, platelet and lymphocyte, and was calculated by neutrophil count × platelet count/lymphocyte count. Some studies had confirmed its prognostic values in malignant tumors, such as cervical cancer (*Huang et al., 2019*), lung cancer (*Hong et al., 2015*), lymphoma (*Wang et al., 2021*), and hepatocellular carcinoma (*Xin et al., 2021*). Other studies demonstrated that the elevated SII might be associated with ulcerative colitis (*Xie et al., 2021*) and predict poor clinical outcome in patients with coronary artery disease (*Yang et al., 2020*).

However, the relationship between SII and DN has not been clearly defined. Therefore, the aim of our study was to explore the relationship between SII and DN. We assumed that an increased SII would be associated with a higher risk of DN among the participants with diabetes mellitus.

## RESEARCH DESIGN AND METHODS

### Study population

We retrospectively investigated 200 patients with type 2 diabetes mellitus between January 2015 and December 2020 at our hospital. Among them, 108 patients were proven to be DN after undergoing renal biopsy, and were included in the DN group, another 92 diabetes patients were included in the DM group because they had no signs of renal injury such as the urinary albumin/creatinine ratio (UACR) >30 mg/g and/or the estimated glomerular filtration rate (eGFR)<60 ml/min/1.73 m2. Exclusion criteria in our study were (1) aged <18 years old, (2) acute infection, (3) malignant tumor, (4) treatment with glucocorticoids and immunosuppressants for autoimmune diseases or connective tissue diseases, (5) complications of hematological disease, (6) complications of liver disease, and (7) missing complete data. All included patients had signed a patient informed consent form at the time of admission. We did not access to information that could identify individual participants during or after data collection. The protocol of this study was approved by the Ethics Review Form for Branch for Medical Research and Clinical Technology Application, Ethics Committee of the First Affiliated Hospital of Fujian Medical University (approval number [2015]084-2).

### Clinical and laboratory data

We selected the blood routine test including white blood cell (10ˆ9/L), neutrophil (10ˆ9/L), lymphocyte (10ˆ9/L), monocyte (10ˆ9/L), platelet (10ˆ9/L) and hemoglobin (g/L). Variables
that may affect the association between SII and DN were included in our study, including age (years), gender (male/female), DM duration(years), diabetic retinopathy(yes/no), use of insulin(yes/no), haemoglobin (HGB, g/L), serum creatinine (SCr, umol/L), blood urea nitrogen (BUN, mmol/L), serum uric acid (UA, mmol/L), serum albumin (Alb, g/L), total cholesterol (mmol/L), Fibrinogen (Fn, g/L), CRP(mg/L), PCT (ng/ml), urinary albumin to creatinine ratio (ACR, mg/g), grade of hematuria (0-3), urine red blood cell (RBC) count(/ul), HbA1 (%) and estimated-glomerular filtration rate (eGFR, ml/min/1.73 m$^2$). The eGFR was estimated by using CKD-EPI equation (*Levey et al., 2009*). SII was calculated as neutrophil count × platelet count/lymphocyte count.

## Statistical analysis

We used the Shapiro–Wilk test to assess whether the data were normally distributed for continuous variables. Data were expressed as means ± standard deviation if they followed a normal distribution. Comparisons between groups were made using the Student's *t*-test. Data were expressed as medians and interquartile ranges if they were not normally distributed. Comparisons between groups were made using the Kruskal-Wallis test. Chi-squared test was used to test for significant differences between groups for categorical variables. The Spearman correlation analysis was conducted between SII and four variables (GFR, SCr, ACR, Alb). ROC curve analysis was performed to assess the value of SII in diagnosis of DN. The area under the curve (AUC) was calculated to evaluate the diagnosis power of SII. The Hosmer-Lemeshow test was used to evaluate goodness of fit. Collinearity statistics was used among codependent variables before using them in a multivariate analysis. Univariate and multivariate logistic analysis were conducted to assess SII as the independently risk factor for DN. A restricted cubic spline model was carried out to show the nonlinear association between SII and DN. Missing values were input by multiple interpolation for continuous variables. Statistical significance was defined as $P<0.05$, and all analysis were performed using R software, version 4.1.2 (*R Core Team, 2021*) and SPSS software, version 25.0 (IBM, Armonk, NY, USA).

# RESULTS

## Baseline characteristics of participants

A total of 200 participants were enrolled, of whom 64% (128) were male, with an average age of 56.6 ± 13.4 years; 54% (108) participants were categorized as DN. According to the complication of renal injury, we divided 200 patients into two groups: DM group and DN group. The clinical and biochemical characteristics of the participants between DM and DN are shown in Table 1. As we can see, except gender, DM duration, use of insulin, other variables were significant different in DM and DN group. In DN group, patients might had higher level of SII, diabetic retinopathy, serum creatinine, blood urea nitrogen, serum uric acid, total cholesterol, white blood cell, neutrophil, lymphocyte, monocyte, platelet, urine RBC count, urinary ACR, fibrinogen, CRP, PCT, and lower level of eGFR, serum albumin, hemoglobin, HbA1c, suggesting that DN group had more severe indicators when compared with DM group.

**Table 1  Clinical characteristics and laboratory tests of patients with DM and DN.**

| Parameter | DM ($n = 92$) | DN ($n = 108$) | $t/ \chi^2 /Z$ | P value |
|---|---|---|---|---|
| SII (10^9/L) | 433.5 (303.2, 619.8) | 767.0 (502.1, 963.9) | −6.349 | <0.001 |
| Age (years) | 61.0 (50.3, 69.8) | 57.0 (48.0, 63.0) | −2.529 | 0.011 |
| Gender (male, %) | 53 (57.6%) | 75 (69.4%) | 3.021 | 0.082 |
| DM duration (years) | 8.50 (2.00, 14.00) | 10.00 (4.00, 13.00) | 0.787 | 0.432 |
| Diabetic Retinopathy (N, %) | 15 (16.3%) | 40 (37.0%) | 10.71 | 0.001 |
| Use of insulin (N, %) | 48 (52.2%) | 65 (60.2%) | 1.297 | 0.255 |
| Grade of hematuria (N, %) | | | 66.623 | <0.001 |
| Negative | 78 (84.8%) | 30 (27.8%) | | |
| Grade 1 | 11 (12.0%) | 45 (41.7%) | | |
| Grade 2 | 2 (2.2%) | 30 (27.8%) | | |
| Grade 3 | 1 (1.1%) | 3 (2.8%) | | |
| eGFR (ml/min/1.73m$^2$) | 103.1 (88.2, 119.4) | 51.2 (39.7, 73.3) | −10.158 | <0.001 |
| Serum albumin (g/L) | 40.9 (38.0, 43.6) | 29.3 (24.8, 34.9) | −9.863 | <0.001 |
| Serum creatinine (μmol/L) | 55.3 (45.1, 67.8) | 129.0 (95.9, 173.3) | −10.940 | <0.001 |
| Blood urea nitrogen (mmol/L) | 4.67 (3.88, 6.26) | 8.74 (6.53, 14.23) | −9.300 | <0.001 |
| Serum uric acid (mmol/L) | 280.2 (236.3, 357.6) | 381.0 (324.8, 452.5) | −6.370 | <0.001 |
| HbA1c (%) | 8.85 (7.25, 11.35) | 7.45 (6.30, 10.91) | −4.268 | <0.001 |
| HbA1c (mmol/mol) | 73 (56,101) | 58 (45,96) | −4.268 | <0.001 |
| Total cholesterol (mmol/L) | 4.51 (3.81, 5.44) | 5.29 (4.00, 6.82) | −3.297 | 0.001 |
| White blood cell (10^9/L) | 5.98 (5.05, 7.20) | 6.88 (5.72, 8.23) | −3.130 | <0.001 |
| Neutrophil (10^9/L) | 3.44 (2.78, 4.51) | 4.63 (3.80, 5.60) | −5.370 | <0.001 |
| Lymphocyte (10^9/L) | 1.76 (1.45, 2.16) | 1.48 (1.15, 1.88) | −3.574 | <0.001 |
| Monocyte (10^9/L) | 0.32 (0.26, 0.42) | 0.40 (0.29, 0.48) | −3.069 | 0.002 |
| Platelet (10^9/L) | 226.1 ± 58.7 | 251.9 ± 78.9 | −2.654 | 0.009 |
| Hemoglobin (g/L) | 133.5 ± 17.1 | 104.1 ± 21.3 | 10.638 | <0.001 |
| Urine RBC count (10^12/L) | 4.45 (2.15, 8.30) | 16.10 (5.73, 48.15) | 6.340 | <0.001 |
| urinary ACR (mg/g) | 9.21 (5.44, 22.93) | 3,703.81 (2,065.37, 7,592.54) | −11.085 | <0.001 |
| Fibrinogen (g/L) | 3.75 ± 1.48 | 4.71 ± 1.25 | −5.013 | <0.001 |
| CRP (mg/L) | 4.66 (2.57, 6.13) | 5.00 (5.00, 6.53) | 2.790 | 0.005 |
| PCT (ng/mL) | 0.01 (0.01, 0.01) | 0.06 (0.01, 1.10) | 4.741 | <0.001 |

**Notes.**
ACR, albumin to creatinine ratio; DM, diabetes mellitus; DN, diabetic nephropathy; Urine RBC, Urine red blood cell; CRP, C-reactive protein; PCT, procalcitonin.
Data are presented as the mean ± standard deviation, or medians (Q1, Q3) or $n$ (%).

## The association between SII and renal function indicators

Using Spearman association analysis to calculate the association between SII and renal function indicators, we found that SII was negative associated with eGFR ($r = −0.368$, $P<0.001$, Fig. 1A) and serum albumin ($r = −0.427$, $P<0.001$, Fig. 1C), while positive associated with increased urinary albumin to creatinine ratio ($r = 0.433$, $P<0.001$, Fig. 1D) and serum creatinine($r = 0.338$, $P<0.001$, Fig. 1B). Patients with higher SII tended to have higher level of urinary albumin to creatinine ratio and serum creatinine, but lower level of eGFR and serum albumin.

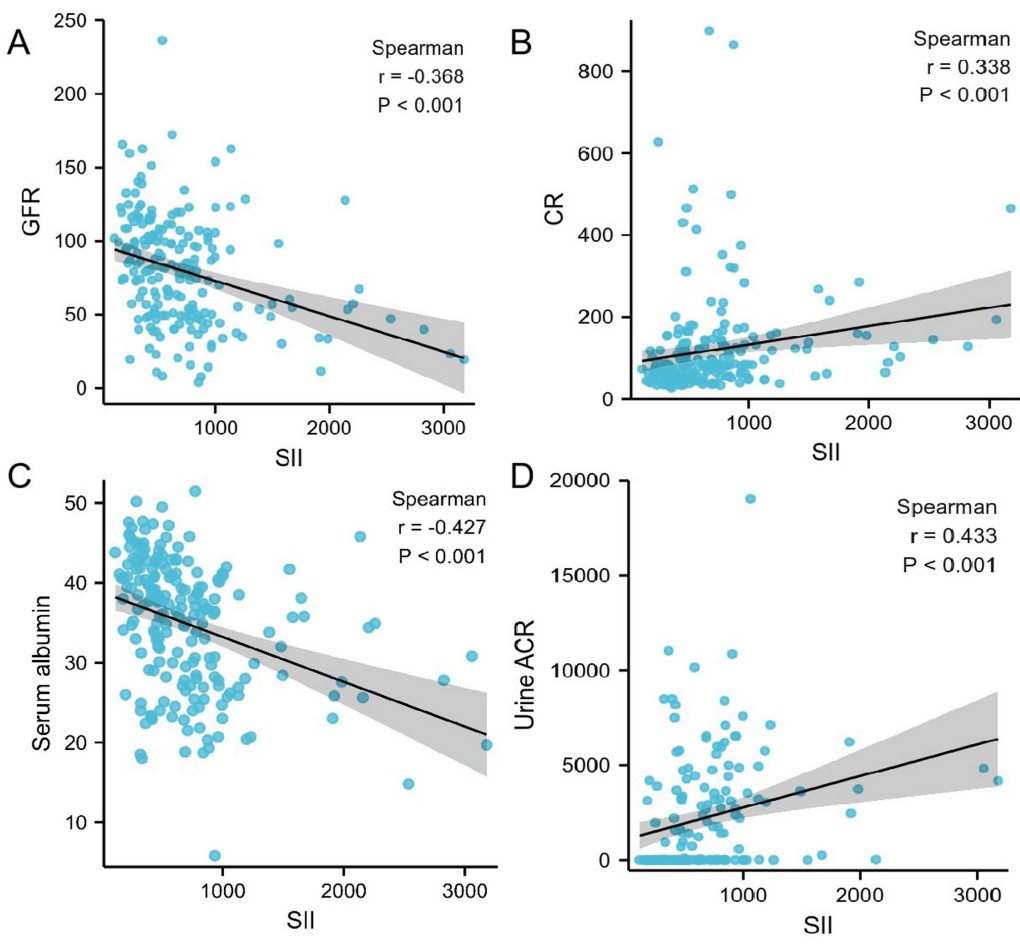

**Figure 1** (A–D) Correalation between SII and different variables calculated by Spearman association analysis.

## ROC curve analysis

The receiver operating characteristic (ROC) curve was performed to assess the value of SII in diagnosis of DN. The maximum area under the curve AUROC was 0.761 (95% *CI*, 0.694−0.828, *P*<0.001), and the cut-off value for diagnosing DN was 624.35*$10^9$/L, the sensitivity was 0.761 and the specificity was 0.667 (Fig. 2).

## Univariate and multivariate logistics regression

To explore the risk factor of developing DN, we used univariate and multivariate logistics regression analysis. In the univariate logistics regression analysis, we found that SII, age, diabetic retinopathy, grade of hematuria, eGFR, serum albumin, serum creatinine, blood urea nitrogen, serum uric acid, total cholesterol, hemoglobin, urine RBC count and fibrinogen were respectively the risk factors of developing DN.

Furtherly, we conducted multivariate logistics regression. We did the collinearity statistics among codependent variables (uric acid, creatinine, eGFR and blood urea nitrogen), with VIF <5, it can be assumed that there is no collinearity between these

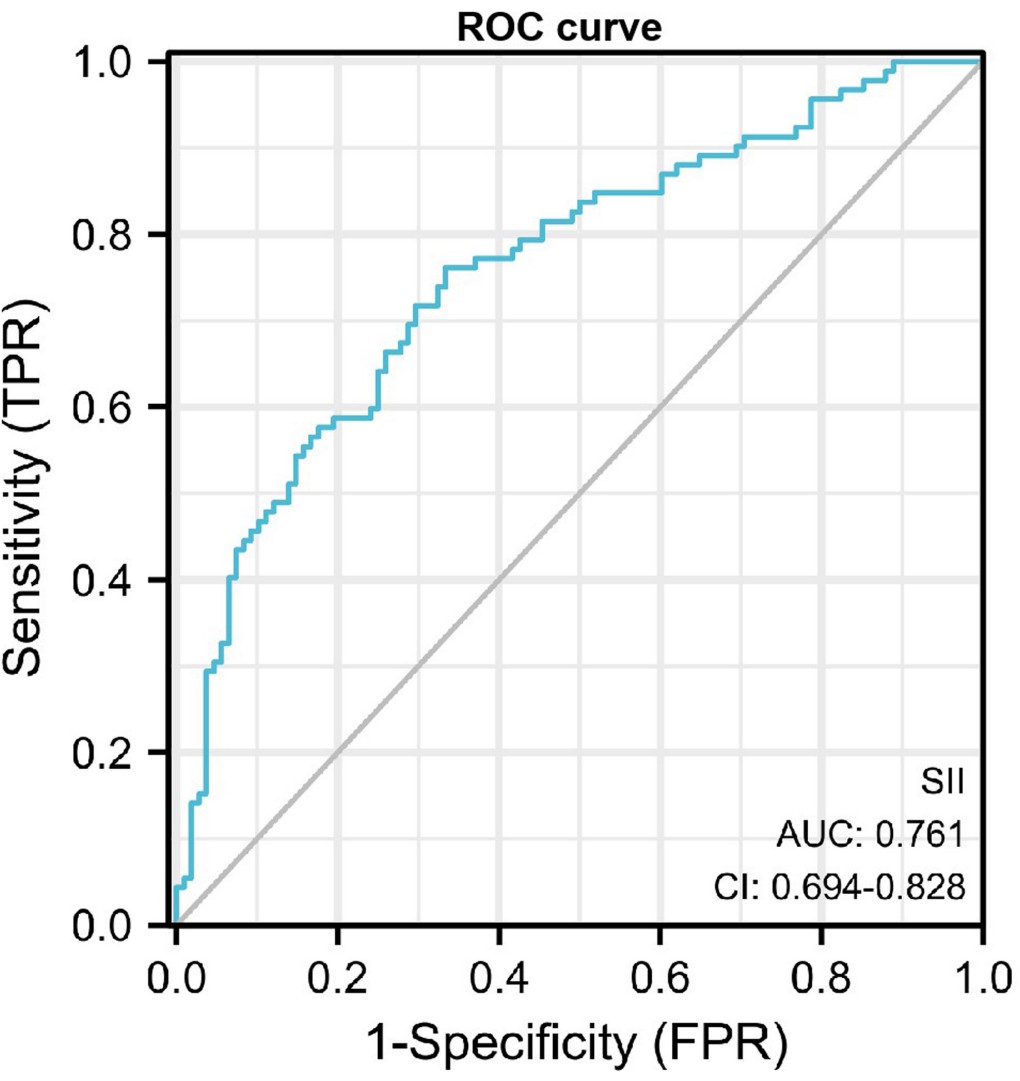

**Figure 2** ROC curve of SII in diagnosis of DN.

variables (Table S3). We included variables that were significant in the univariate analysis such as age, diabetic retinopathy, grade of hematuria, urine RBC count, serum creatinine, serum uric acid, blood urea nitrogen, hemoglobin, serum albumin, total cholesterol, fibrinogen and eGFR were included in the adjusted model for the multivariate logistics regression. We used forward wald method for the multivariate analysis. The Hosmer-Lemeshow test was used to evaluate goodness of fit. The Hosmer-Lemeshow test suggested that the final model $p$-value was 0.855, which was greater than 0.05 and was considered to be a good fit (Table S4). Finally, SII ($OR = 1.004$, 95% $CI$: $1.001-1.006$, $P = 0.002$) and serum creatinine ($OR = 1.146$, 95% $CI$: $1.069-1.228$, $P<0.001$) were independently risk factors for the occurrence of DN, while age ($OR = 0.920$, 95% $CI$: $0.863-0.981$, $P = 0.011$) and serum albumin ($OR = 0.708$, 95% $CI$: $0.595-0.844$, $P<0.001$) were protective factors for the occurrence of DN (Table 2).

**Table 2 Univariate and multivariate logistics regression for DN.** *P* values are shown in bold.

| Variables | Univariate | | | Multivariate | | |
|---|---|---|---|---|---|---|
| | *OR* | *95% CI* | *P* value | *OR* | *95% CI* | *P* value |
| SII | 1.003 | 1.002, 1.004 | <0.001 | 1.004 | 1.001, 1.006 | **0.002** |
| Age | 0.975 | 0.954, 0.996 | 0.022 | 0.920 | 0.863, 0.981 | **0.011** |
| Gender | 1.672 | 0.935, 2.993 | 0.083 | | | |
| DM duration | 1.017 | 0.977, 1.059 | 0.410 | | | |
| Diabetic Retinopathy | 3.020 | 1.534, 5.943 | 0.001 | | | |
| Use of insulin | 1.386 | 0.790, 2.431 | 0.255 | | | |
| Grade of hematuria | | | | | | |
| Grade 1 | 10.636 | 4.865, 23.255 | <0.001 | | | |
| Grade 2 | 39.000 | 8.772, 173.394 | <0.001 | | | |
| Grade 3 | 7.800 | 0.780, 77.955 | <0.001 | | | |
| eGFR | 0.926 | 0.908, 0.945 | <0.001 | | | |
| Serum creatinine | 1.099 | 1.067, 1.132 | <0.001 | 1.146 | 1.069, 1.228 | **<0.001** |
| Blood urea nitrogen | 2.019 | 1.621, 2.514 | <0.001 | | | |
| Serum uric acid | 1.010 | 1.006, 1.013 | <0.001 | | | |
| Hemoglobin | 0.925 | 0.905, 0.945 | <0.001 | | | |
| Urine RBC count | 1.020 | 1.008, 1.033 | 0.001 | | | |
| Serum albumin | 0.736 | 0.676, 0.801 | <0.001 | 0.708 | 0.595, 0.844 | **<0.001** |
| Total cholesterol | 1.382 | 1.156, 1.654 | <0.001 | | | |
| HbA1c | 0.952 | 0.870, 1.042 | 0.284 | | | |
| Fibrinogen | 1.700 | 1.349, 2.143 | <0.001 | | | |
| CRP | 1.003 | 0969, 1.039 | 0.854 | | | |
| PCT | 1.081 | 0.927, 1.259 | 0.321 | | | |

## The restricted cubic spline model

From the restricted cubic spline model, we can see SII was independently associated with the occurrence of DN. The restricted cubic spline was plotted using five default knots. The *P*-value for the nonlinear association was 0.047 (Fig. 3).

## DISCUSSION

Our finding suggested that higher SII level was an independent risk factor for the increased likelihood of DN. Previous studies have reported the association between SII and increased risk of renal injury, such as albuminuria (*Qin et al., 2022*) and acute kidney injure in the acute pancreatitis (*Lu et al., 2022*). Consistent with most studies, our study demonstrated that higher SII level might be independently associated with DN.

SII was calculated as neutrophil count × platelet count/lymphocyte count. As we know, neutrophils induce and activate inflammatory response, and lymphocytes can partly reflect the immune statue. DN had been considered as an inflammatory disease characterized by leukocyte infiltration at almost every stage of renal involvement (*Galkina & Ley, 2006*). Platelet activating factor mediates the inflammatory effect of platelet-neutrophils, which then induces the aggregation, produce reactive oxygen species and infiltration of

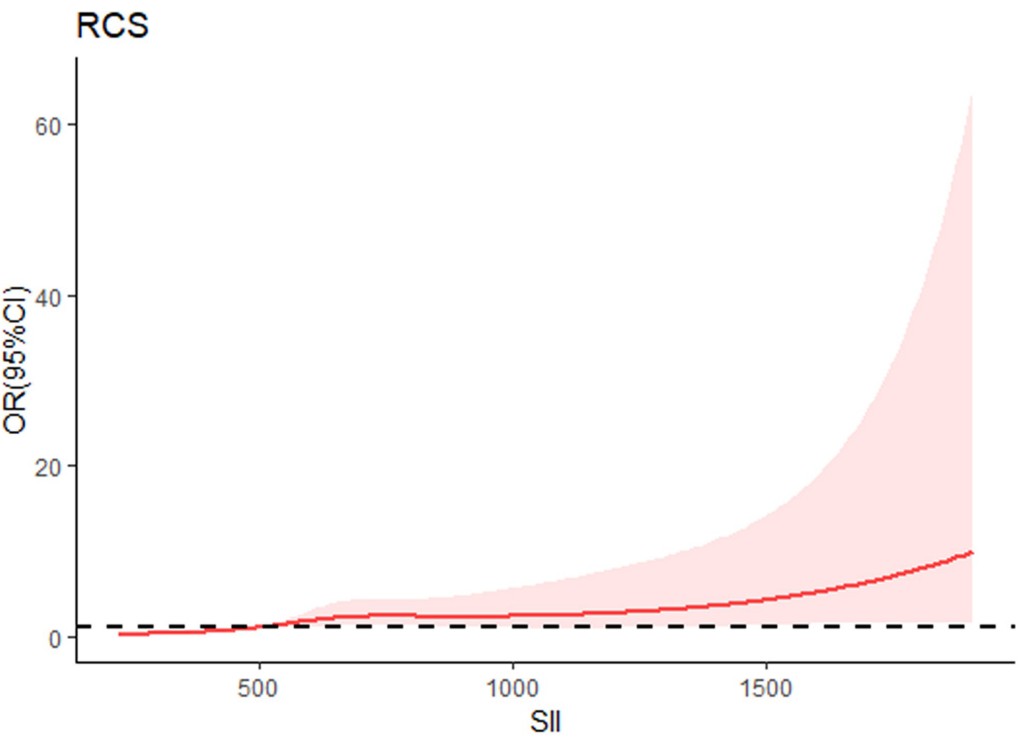

**Figure 3 Continuous association of SII with the incidence of DN by a restricted cubic spline.** The restricted cubic spline was plotted using five default knots. The *P*-value for the nonlinear association was 0.047.

inflammatory cells, further amplifying the inflammatory response, so increased number of platelets can indirectly reflect inflammatory state (*Van der Meijden & Heemskerk, 2019*). A decline in lymphocytes predicts cellular immune injury, and increased numbers of neutrophils and platelets are considered responses to systemic inflammation. A higher SII indicates greater inflammation and a weaker immune response.

There is growing evidence for inflammation in modulating the process of DN (*Hung et al., 2021*; *Kelly & Dominguez, 2010*; *Matoba et al., 2019*). Inflammation and inflammatory molecules may affect renal function by altering vascular permeability, affecting vasodilatory and contractile functions, regulating the proliferation of mesangial cells, endothelial cells and vascular smooth muscle cells, and inducing cytotoxicity, apoptosis and necrosis (*Fufaa et al., 2016*). Recent new advancements in discovery of DN biomarkers can help us to identify the disease in an early stage and to improve risk stratification (*Barutta et al., 2021*). In DN patients, leukocytes are activated by advanced glycation end products (AGEs), oxidative stress and cytokines to secrete a variety of cytokines and accelerate the DN process (*Navarro-González et al., 2011*; *Samsu, 2021*). Chronic hyperglycemia was found to cause an increase in daily basal reactive oxygen species (ROS) production of neutrophils (*Saito et al., 2013*). Macrophages, neutrophils, and T lymphocytes produce various reactive oxygen species, growth factors and proinflammatory cytokines, which modulate the local response and increase inflammation in DN (*Galkina & Ley, 2006*). In
a study of American Indian patients with T2DM, it was revealed that a lower lymphocyte count was associated with the development of DN. In addition, long-term hyperglycemia increases the apoptosis of lymphocytes, the total number of lymphocytes decreases, resulting in sustained inflammation statue, promoting oxidative stress, leading to renal tubulointerstitial lesions, and accelerating the progression of renal fibrosis (*Eller et al., 2011*). A study found that higher levels of proinflammatory monocytes and circulatory inflammatory mediators existed in DN patients, suggesting the importance of inflammation and endothelial dysfunction in DN (*Kolseth et al., 2017*). Another study showed that markers of inflammation (including high-sensitivity C-reactive protein, tumor necrosis factor-$\alpha$ receptor 2, white blood cell count, and interleukin-6) predict the long-term risk of developing chronic kidney disease (*Shankar et al., 2011*). Maybe these are the potential reasons why higher level of SII is associated with the development of DN.

SII showed a positive correlation with ACR and serum creatinine level. Thus, our findings revealed that increased SII might indicate the risk of kidney injury in diabetic patients. To evaluate the predictive value of SII for DN risk, a ROC curve analysis was carried out. The AUC for SII was 0.761, suggesting that SII had moderate predictive power for DN. Furthermore, based on a cutoff value of 624.35, SII exhibited moderate sensitivity and specificity, diagnostic accuracy of 76.1% and 66.7% respectively, for prediction for DN. Therefore, the results showed that, as an independent risk factor, SII had some extent predictive power for DN. The sensitivity and specificity are also not particularly high, indicating that SII alone may not be sufficient as a diagnostic tool without additional markers or clinical factors. Therefore, in future studies, we intend to develop combinations of other markers to improve the predictive value of DN.

When continuous variables are nonlinearly related to an outcome, the use of RCS curves provides a more accurate description of the risk relationship between continuous variables and the occurrence of the outcome. Therefore, we used RCS analysis to further investigate the nonlinear association between SII and the risk of developing DN. Results of RCS analysis revealed that the SII was positively correlated with the incidence of DN in a nonlinear pattern (P for nonlinear = 0.047). The risk of DN increased with increasing SII, especially if the SII was greater than 624*10^9/L. Therefore, it is essential to control SII in a certain range, and our results may provide new evidences for clinicians to prevent DN.

To determine whether SII could be used as a risk factor for DN, univariate and multivariate logistics regression analyses were performed. When we conducted multivariate logistics regression model adjusted for clinical covariates, we discovered that SII was a still independently risk factor for DN. We also observed the same conclusion in another study, which included 3,937 people from NHANES database (*Guo et al., 2022*). Although the number of patients included was small, the DN patients in our study were diagnosed through renal biopsy, which further verified the conclusions in the real world. Therefore, it is of great value to explore the significance of the SII for DN risk assessment and prediction.

However, there were several limitations existed in current study. First of all, it was a single center, retrospective study with limited sample size, suggesting that there might have been selection bias and limiting the applicability of the results to broader populations. Our study also lacked an external validation cohort. In the future studies, we will include

multicenter prospective studies and more diverse population to reduce selection bias and improve the robustness of the results and enhance the generalizability of the findings. Additionally, SII is the ratio of blood cell parameters, and blood cells can be affected by a variety of factors, such as infection, dehydration, excessive water in the body, and dilution of blood specimens, and other systemic conditions, so it might be not reliable enough to use them alone in clinical diagnosis and prediction of diseases. Only for patients without underlying diseases, immune system diseases and obvious systemic or local infections, SII might has some diagnostic value. Future studies should include a prospective study design or an external validation cohort to strengthen the evidence for SII as an independent risk factor for DN. Futhermore, in future studies, we will explore other inflammatory or immune-related biomarkers in combination with SII to improve predictive accuracy and clinical utility. Larger-scale, prospective studies are needed to further confirm our findings.

## CONCLUSIONS

In conclusion, our study demonstrated that elevated SII level was some extent associated with DN. SII is a novel diagnostic biomarkers and universally available method characterized by a non-invasive approach, easy access and low cost, and it thus has some promising prospects for application in predicting DN. Further large-scale prospective studies are still needed to validate our findings before SII can be considered a reliable diagnostic or prognostic tool for DN.

### Funding

This work was supported by the Natural Science Foundation of Fujian Province under Grant (No. 2022J01212) and the Fujian Provincial Health Technology Project under Grant (No. 2021CXA018). The funders had no role in study design, data collection and analysis, decision to publish, or preparation of the manuscript.

### Grant Disclosures

The following grant information was disclosed by the authors:
The Natural Science Foundation of Fujian Province: 2022J01212.
the Fujian Provincial Health Technology Project: 2021CXA018.

### Competing Interests

The authors declare there are no competing interests.

### Author Contributions

- Xiaohong Zhang conceived and designed the experiments, performed the experiments, prepared figures and/or tables, authored or reviewed drafts of the article, and approved the final draft.

- Yuan Fang conceived and designed the experiments, performed the experiments, analyzed the data, prepared figures and/or tables, authored or reviewed drafts of the article, and approved the final draft.
- Mengjie Weng conceived and designed the experiments, performed the experiments, authored or reviewed drafts of the article, and approved the final draft.
- Caiming Chen conceived and designed the experiments, performed the experiments, authored or reviewed drafts of the article, and approved the final draft.
- Yanfang Xu conceived and designed the experiments, authored or reviewed drafts of the article, and approved the final draft.
- Jianxin Wan conceived and designed the experiments, authored or reviewed drafts of the article, and approved the final draft.

## Human Ethics

The following information was supplied relating to ethical approvals (*i.e.*, approving body and any reference numbers):

The study adhered to all relevant tenets of the Declaration of Helsinki and was approved by the Ethics Committee of the First Affiliated Hospital of Fujian Medical University (MTCA, ECFAH of FMU [2015] 084-2).

## Ethics

The following information was supplied relating to ethical approvals (*i.e.*, approving body and any reference numbers):

The study adhered to all relevant tenets of the Declaration of Helsinki and was approved by the Ethics Committee of the First Affiliated Hospital of Fujian Medical University (MTCA, ECFAH of FMU [2015] 084-2).

## Data Availability

The raw data are available in the Supplemental Files.

## Supplemental Information

Supplemental information for this article can be found online at http://dx.doi.org/10.7717/peerj.18493#supplemental-information.

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
