# Peer review of "Systemic immune-inflammation index as an independent risk factor for diabetic nephropathy: a retrospective, single-center study"

_PeerJ, doi:10.7717/peerj.18493_

## Round 0.1 · original submission · Major Revisions

Many confounding variables were not analysed and the model of regression analysis did not resolved the collinearity. Therefore, the manuscript needs some major changes.

Reviewer 1 ·

Basic reporting

1. Relevance of the Study: The study addresses an important area of research, given the significant burden of diabetic kidney disease (DKD) worldwide. The exploration of the systemic immune-inflammation index (SII) as a potential biomarker for DKD is innovative and clinically relevant.
2. Methodological Rigor: The use of a restricted cubic spline model to explore the nonlinear association between SII and DKD adds robustness to the analysis. Additionally, the multivariate logistic regression model helps in identifying independent risk factors for DKD, accounting for potential confounders.
3. Statistical Analysis: The manuscript employs a comprehensive range of statistical methods, including Spearman correlation, ROC curve analysis, and logistic regression, which are appropriate for the study design and objectives.

Experimental design

1. Sample Size and Generalizability: The study is based on a relatively small sample size (n=200), which limits the generalizability of the findings. The single-center, retrospective nature of the study also raises concerns about potential selection bias and limits the applicability of the results to broader populations.
2. Lack of External Validation: The study lacks an external validation cohort, which would have strengthened the findings. The results might have been more compelling if validated in a separate, larger population or in a prospective study.
3. Confounding Factors: Although the study adjusts for several confounders, there is a possibility that other important factors influencing SII and DKD (e.g., duration of diabetes, glycemic control, and other inflammatory markers) were not adequately controlled. This could have impacted the observed associations.

Validity of the findings

1. Discussion and Interpretation: The discussion section could benefit from a more critical interpretation of the results, particularly regarding the limitations of using SII as a standalone biomarker. The authors should discuss the potential limitations of SII in clinical practice, including its variability due to factors like infection, dehydration, and other systemic conditions.
2. ROC Curve Analysis: While the ROC curve analysis shows that SII has predictive value for DKD, the reported AUC (0.761) suggests only moderate discriminative ability. The sensitivity and specificity are also not particularly high, indicating that SII alone may not be sufficient as a diagnostic tool without additional markers or clinical factors.

Additional comments

Suggestions for Improvement:
1. Expand Sample Size and Study Population: Future studies should include a larger and more diverse population to enhance the generalizability of the findings. Multicenter studies could help reduce selection bias and improve the robustness of the results.
2. Incorporate Prospective Validation: Incorporating a prospective study design or an external validation cohort would strengthen the evidence for SII as an independent risk factor for DKD.
3. Explore Additional Biomarkers: Consider exploring other inflammatory or immune-related biomarkers in combination with SII to improve predictive accuracy and clinical utility.
4. Enhance Discussion on Clinical Implications: The manuscript would benefit from a more nuanced discussion of how SII could be integrated into clinical practice, including potential challenges and the need for further validation.
5. Refine Conclusion: The conclusion should be more tempered, emphasizing the need for further research before SII can be considered a reliable diagnostic or prognostic tool for DKD.

Reviewer 2 ·

Basic reporting

1. The term ‘Diabetic kidney disease’ (DKD) is defined by the coexistence of CKD with diabetes mellitus, regardless of the histologic diagnosis (if it exists). Therefore, considering that in the current study the histologic diagnoses are known, diabetic nephropathy (supposing this was the actual diagnosis and not other nephropathy) is probably more suitable to be used instead of DKD.
2. The phrase from lines 69-71 should be rephrased since it seems to be missing a word.
3. In line 91 the word 'were' is written 2 times.

Experimental design

1. The authors defined the DM group by the absence of proteinuria (how was it measured? what about albuminuria?, what level of albuminuria was considered normal?) and renal dysfunction (based on what definition?). According to KDIGO guidelines, proteinuria and creatinine levels (kidney function) are not the only markers for kidney lesions and chronic kidney disease. Therefor, from the manuscript is not clear whether this group had any other markers indicating kidney disease (that can manifest with no proteinuria and normal kidney function, such as hematuria).

2. For the "DKD group", it would be useful to know the reason for kidney biopsy in this patients, since not all the patients with DM and indicators of kidney disease undergo kidney biopsy.

3.The proposed regression model has three main problems:
a. first, it uses codependent variables (uric acid, creatinine, eGFR and blood urea nitrogen) - the authors should check for collinearity among variables before using them in a multivariate analysis
b. secondly, the authors do not include in the multivariate analysis variables that are far more important and could explain the development of diabetic nephropathy. According to current literature, commonly reported predictors for diabetic nephropathy are longer duration of DM, higher glycosylated hemoglobin, presence of retinopathy, higher blood pressure, proteinuria, lower hemoglobin, higher serum creatinine, and insulin treatment. Therefor, considering that these important information was omitted from the regression, the results are not valid.
c. thirdly, when reporting the results of the logistic regression, i recommend to report the used method for the multivariate analysis (enter, backward wald, forward wald etc) and the values of goodness of fit tests.

4. The reason and clinical utility of the restricted cubic spine model is not clear. I recommend providing more information about this analysis.

Validity of the findings

The research idea is for sure of interest, however, the main weakness of the study is the missing of very important variables that are related to diabetic nephropathy (see above). In this context, the main findings and conclusions of the study can not be interpreted as valid. I recommend resolving this issues before resubmitting.

---

## Round 0.2 · Major Revisions

I consider that the changes made in response to the reviewers' suggestions are not enough. Major concerns related to the methodology (selection of the subjects, definitions of the groups, terminology, type of the study design and statistical analysis) were not resolved. Therefore, in the current form the manuscript is not suitable for publication. If the authors are willing to make profound changes in accordance with the reviewers' comments (especially reviewer 2), a secondly revised version could be taken into account.

Reviewer 1 ·

Basic reporting

Thank you for submitting your revised manuscript. Below are my comments and suggestions for further improvement:
1. Sample Size and Generalizability:
o Your acknowledgment of the limitations regarding sample size and the single-center nature of the study is well noted. The addition of prospective, multicenter studies in future research will certainly improve the robustness of the results. The revisions you made to address this concern in the discussion section are appropriate and clearly stated.
2. Lack of External Validation:
o I appreciate the clarification that no external validation cohort was available in this study. Your commitment to validating the findings in future prospective studies is important and strengthens the value of your research. The revision in the discussion section reflects this well.
3. Confounding Factors:
o Including additional confounding factors like the duration of diabetes and inflammatory markers such as CRP and PCT is commendable. The re-analysis performed reassures that the adjustments did not alter the overall findings. Your revision effectively addresses this concern.

Experimental design

4. Validity of Findings - SII as a Biomarker:
o You have appropriately acknowledged the limitations of using SII as a standalone biomarker and expanded on the potential influences of various systemic conditions. This addition improves the manuscript by offering a more nuanced view of the limitations of SII in clinical practice.
5. ROC Curve Analysis:
o The interpretation of the ROC curve analysis is sound, and I appreciate your plans for future studies that explore combinations of other biomarkers to enhance predictive accuracy. The revision in the discussion section reflects a balanced view of the current findings and future directions.

Validity of the findings

4. Validity of Findings - SII as a Biomarker:
o You have appropriately acknowledged the limitations of using SII as a standalone biomarker and expanded on the potential influences of various systemic conditions. This addition improves the manuscript by offering a more nuanced view of the limitations of SII in clinical practice.
5. ROC Curve Analysis:
o The interpretation of the ROC curve analysis is sound, and I appreciate your plans for future studies that explore combinations of other biomarkers to enhance predictive accuracy. The revision in the discussion section reflects a balanced view of the current findings and future directions.

Additional comments

8. Conclusion:
o Your revised conclusion is more measured and appropriately emphasizes the need for further validation of SII before it can be considered a reliable diagnostic or prognostic tool for DKD. The tempered tone in the conclusion enhances the scientific rigor of the manuscript.
Overall Recommendation:
The revisions made have significantly improved the clarity and robustness of the manuscript. However, I recommend minor further refinements in the discussion section to strengthen the emphasis on the limitations and the need for larger-scale, prospective studies to confirm the findings. Once addressed, the manuscript will be suitable for publication.

Reviewer 2 ·

Basic reporting

I appreciate the effort to address the previously raised issues, however, there is a major issue that has not been addressed - the inappropriate use of the term Diabetic kidney disease. This is important for the study main question and results, and you will find my arguments below.
My initial comment was:
The term ‘Diabetic kidney disease’ (DKD) is defined by the coexistence of CKD with diabetes mellitus, regardless of the histologic diagnosis (if it exists). Therefore, considering that in the current study the histologic diagnoses are known, diabetic nephropathy (supposing this was the actual diagnosis and not other nephropathy) is probably more suitable to be used instead of DKD.
Response: We thank the reviewer’s suggestion.
In 2007, National Kidney Foundation⁃Kidney Disease Outcome Quality Initiative (NKF⁃KDOQI)firstly suggested to adopt the concept of ‘DKD’ to replace ‘diabetic nephropathy (DN)’. In 2014, American Diabetes Association (ADA) and NKF reached a consensus on the concept of ‘DKD’, stating that ‘DKD’ is CKD caused by DM, although the name ‘DN’ still remains in the domestic and international nephrology circles,the concept of ‘DKD’ is gradually replacing ‘DN’,therefore, we use “DKD” in our study.

My response: With all due respect for the work of the authors, I disagree with the affirmation that the authors made that ”DKD is CAUSED by DM”. As a matter of fact, in the newer KDIGO 2022 Clinical Practice Guideline for Diabetes Management in Chronic Kidney Disease you will find the following sentence „We avoid the use of the term “diabetic kidney disease” to avoid the connotation that CKD is caused by traditional diabetes pathophysiology in all cases.” However, analyzing the current literature, DKD is defined by the COEXISTENCE of CKD and diabetes mellitus and there may be no causality between the two entities since DKD comprises those patients diagnosed by kidney biopsy with diabetic nephropathy lesions (when only specific diabetic lesions are seen) and non-diabetic kidney lesions (when the lesions are unrelated to diabetes - such as any other glomerulopathy, or a vascular nephropathy or tubulo-interstitial nephropathy), or a combination of the two. From the literature you will find out that the percentage of non-diabetic kidney disease in subjects with DM ranges between 30 and 75%. Considering this wide range and the fact that the pathophysiology of diabetic nephropathy is evidently different from other kidney diseases (and therefor the possible contribution of inflammation may differ), it is important for the current manuscript to clearly specify what kind of kidney disease is investigated. Since the authors mention that the diagnosis of DKD was made after undergoing kidney disease, they should clearly mention what kind of histologic lesions they found (how many DN and how many other nephropathies). However, if the authors included in the study both subjects with DN and other non-diabetic lesions, the research question is faulty, and the study should be restricted to only diabetic nephropathy subjects.
For further information and clarification of the subject I recommend the work of Sharma G (10.2215/CJN.02510213), Suneja M (10.4236/jdm.2021.115029 ).

Second of all, the rephrased sentence from the first part of Research design and methods „92 patients were diagnosed as DM because of a urinary albumin/creatinine ratio (UACR)<30mg/g and/or an estimated glomerular filtration rate (eGFR)≥60mL/min/1.73m2” reflects a logical error due to improper naming of the DM group. Evidently, patients are not diagnosed as DM due to urinary changes or modification of eGFR! I recommend changing the name of the respective group and rephrasing.

Experimental design

1. I appreciate adding the data regarding DM duration, insulin treatment, diabetic retinopathy and hematuria. However, these variables were not introduced in the multivariate analysis which was the main purpose – to see if, independent of this important cofounders, SII still remains an independent risk factor for DKD. I recommend the authors to remake a new model that includes those variables that were significant in the univariate analysis.
2. As the authors explained in the manuscript, the restricted cubic spline is a survival analysis, however, for the current raw data, the use of any survival analysis is not appropriate since there is no time to event information – the study is transversal and not longitudinal. Therefor, I am in doubt that the restricted cubic spline analysis is appropriate for the current study. The authors should consult with a statistician. Anyway, there is no sense for the discussion about survival analysis in the discussion sections since this is not a longitudinal study.

Validity of the findings

Considering the previously raised question regarding the statistical analysis, the validity of the finding is questionable.

---

## Round 0.3 · accepted · Accept

No further comments. The revised manuscript is suitable for publication.

Reviewer 1 ·

Basic reporting

The revised manuscript has updated the concerns I raised previously in last two revisions.

Experimental design

My comments are well documented on this section on my previous comments to this manuscript

Validity of the findings

Acceptable

Reviewer 2 ·

Basic reporting

Most of the issues have been addressed. I have no further comments.

Experimental design

Most of the issues have been addressed. I have no further comments.

Validity of the findings

Most of the issues have been addressed. I have no further comments.